# One Stage Masquelets Technique: Evaluation of Different Forms of Membrane Filling with and without Bone Marrow Mononuclear Cells (BMC) in Large Femoral Bone Defects in Rats

**DOI:** 10.3390/cells12091289

**Published:** 2023-04-30

**Authors:** Nicolas Söhling, Myriam Heilani, Charlotte Fremdling, Alexander Schaible, Katrin Schröder, Jan C. Brune, Volker Eras, Christoph Nau, Ingo Marzi, Dirk Henrich, René D. Verboket

**Affiliations:** 1Department of Trauma, Hand and Reconstructive Surgery, Goethe University Frankfurt, 60590 Frankfurt am Main, Germany; myriam.heilani@hotmail.com (M.H.); charlotte.fremdling@gmx.de (C.F.); alexanderschaiblex@gmail.com (A.S.); christoph.nau@kgu.de (C.N.); marzi@trauma.uni-frankfurt.de (I.M.); d.henrich@trauma.uni-frankfurt.de (D.H.); rene.verboket@kgu.de (R.D.V.); 2Center of Physiology, Cardiovascular Physiology, Goethe University Frankfurt, 60590 Frankfurt am Main, Germany; schroeder@vrc.uni-frankfurt.de; 3German Institute for Cell and Tissue Replacement (DIZG, gemeinnützige GmbH), 12555 Berlin, Germany; j_brune@dizg.de (J.C.B.); v_eras@dizg.de (V.E.)

**Keywords:** masquelet, one stage procedure, critical bone defect, BMC

## Abstract

The classic two-stage masquelet technique is an effective procedure for the treatment of large bone defects. Our group recently showed that one surgery could be saved by using a decellularized dermis membrane (DCD, Epiflex, DIZG). In addition, studies with bone substitute materials for defect filling show that it also appears possible to dispense with the removal of syngeneic cancellous bone (SCB), which is fraught with complications. The focus of this work was to clarify whether the SCB can be replaced by the granular demineralized bone matrix (g-DBM) or fibrous demineralized bone matrix (f-DBM) demineralized bone matrix and whether the colonization of the DCD and/or the DBM defect filling with bone marrow mononuclear cells (BMC) can lead to improved bone healing. In 100 Sprague Dawley rats, a critical femoral bone defect 5 mm in length was stabilized with a plate and then encased in DCD. Subsequently, the defect was filled with SCB (control), g-DBM, or f-DBM, with or without BMC. After 8 weeks, the femurs were harvested and subjected to histological, radiological, and biomechanical analysis. The analyses showed the incipient bony bridging of the defect zone in both groups for g-DBM and f-DBM. Stability and bone formation were not affected compared to the control group. The addition of BMCs showed no further improvement in bone healing. In conclusion, DBM offers a new perspective on defect filling; however, the addition of BMC did not lead to better results.

## 1. Introduction

The masquelet technique takes a central position in the treatment of critical bone defects. In the first step of the two-stage surgical procedure, a membrane is induced in the area of the defect zone by a foreign body reaction. For this purpose, a polymethylmethacraylate bone cement plug is inserted into the defect and left in place for 6 weeks. In the subsequent procedure, the PMMA plug is removed, and the cavity surrounded by the membrane is filled with autologous cancellous bone and closed by a suture. The induced membrane serves as a barrier and prevents the ingrowth of connective tissue and the resorption of the inserted bone substitutes and increases the concentration of growth factors by active secretion [1,2]. The formation of pseudarthrosis is presumably prevented. Using this technique, even critically large bone defects can be treated successfully [3,4,5]. Current research is now focused on optimizing this method with the goal of improving bone quality and healing time. Central to this is also the effort to turn the two-stage procedure into a one-stage procedure and reduce surgical trauma. The use of a bone substitute material instead of autologous cancellous bone is the first step. The additional trauma of autologous cancellous bone harvesting (gold standard) with its donor site morbidities could be avoided, and the operation procedure could be accelerated. Alternatives such as allogeneic and mineral bone substitutes are being intensively examined. Above all, the demineralized bone matrix (DBM) has been shown to be superior to mineral substrates in our own work when applied as defect fillings in the femoral critical size defect model of the rat [6]. The positive influence of growth factors released by the implanted DBM is suspected; furthermore, the collagenous structure of the material enables the adhesion of cells [7,8]. However, the volume/surface ratio, the particle size, and the density of the inserted material also play a significant role [9,10,11,12,13]. The results of autologous cancellous bone have not yet been surpassed. Combination experiments of bone graft substitutes with osteogenic cells or stem cells should address this issue. For example, MSC, in combination with β-tricalcium phosphate (TCP), showed a high cytocompatibility and stimulation of bone healing [14,15]. However, the use of MSC is limited by complex and lengthy cultivation in clinical practice. Bone marrow-derived mononuclear cells (BMC)—a heterogeneous population of bone marrow mononuclear cells (containing lymphocytes, monocytes, hematopoietic stem cells (HSC), and endothelial stem cells (EPC), MSC)—are much easier to handle, and also show a high regenerative potential [16]. Seebach et al. demonstrated that the use of BMC in 5 mm critical bone defects in rat femurs (CSD) provided equally good bone healing results as a combination of MSC and EPC [17,18]. In a recent project, bone healing using the same CSD model with two different DBM morphologies as defect fillings in combination with and without BMC colonization was investigated [19]. No significant improvement in bone healing compared to the autologous graft was shown, so further optimization of this technique is necessary. Influencing the induced membrane is another target in the process optimization of the Masquelet technique. The induced membrane itself is usually about 1–2 mm thick, completely interspersed with vessels, and has fibroblasts, myofibroblasts, and collagen, similar to a synovial membrane. Due to the foreign body reaction, inflammatory cells such as macrophages and giant cells have also been found [20]. Membrane formation and cell content can be influenced by additives in bone cement. On the other hand, the maturation time of the membrane or the age of the membrane plays a rather minor role [20,21,22]. However, the most serious disadvantage of the Masquelet technique, the two-stage procedure, remains. For process optimization, a membrane substitute is needed that functions as a selective–permeable barrier and allows cell infiltration, angiogenesis, and tissue regeneration. Although initial in vitro experiments with artificial membranes, such as polycaprolactone, are promising, there is no real alternative to the induced membrane yet [23]. In recent work, our group was able to successfully use the human decellularized dermis (DCD) as a membrane substitute [24]. The initial intervention for membrane establishment could, thus, be saved. Bone healing was shown to be equivalent to that of the induced membrane [24]. In this initial investigation, an unpopulated DCD graft with the syngeneic cancellous bone filling was used. The aim of this work is now to clarify whether the additional loading of the DCD with BMC had a positive effect on bone healing. For this purpose, bone defects in the rat (CSD) were coated with BMC-loaded or cell-free DCD, and the cavity was filled with BMC-loaded or cell-free DBM (in fibrous or granular shape).

## 2. Materials and Methods

### 2.1. Ethics

All animal experiments were performed according to the applicable regulations of the Animal Protection and Monitoring Committee of our institution (project no. FK/1075; Regierungspräsidium, Darmstadt, Germany) in accordance with German law.

### 2.2. Animal Care and Group Setup

For the experiments, 120 8–10-week-old Sprague Dawley rats were used. (Janvier Labs, Saint Berthevin, France). The 250–300 g animals were kept under equal conditions in groups of 3–4 individuals per cage (temperature: 15–21 °C, air circulation, light-controlled with 14 h light and 10 h night per day). The animals had free access to rat food and water. The condition of the animals was checked daily for one week after surgery and weekly thereafter by a veterinarian. For the experiment, the animals were randomized and distributed into 10 groups according to Table 1. Two different defective envelopes were combined with five different defective fillings. The defect fillings used were syngeneic bone (groups 1 and 2), granular DBM (g-DBM) (groups 3 and 4), granular DBM combined with BMC (g-DBM + BMC) (groups 5 and 6), fibrous DBM (f-DBM) (groups 7 and 8) and fibrous DBM combined with BMC (f-DBM + BMC) (groups 9 and 10). The defect envelopes studied were human decellularized dermis and human decellularized dermis colonized with BMC. Twenty animals served as the source for the BMC. DBM granules used (Deutsches Institut für Zell-und Gewebeersatz (DIZG) gemeinnützige GmbH, Berlin, Germany), which were sterilized, according to a validated GMP-compliant procedure and they were approved as a medicinal product according to §21 of the German Drug Law (approval number: PEI.H.03358.01.1). f-DBM was manufactured under controlled computerized numerical control (CNC) milling conditions. The tissue was then partially demineralized and subjected to a validated sterilization process [25]. All DBM products were manufactured from the bones of serologically tested voluntary donors. The manufacturing processes were fully validated, including demineralization, sterilization, and preservation.

### 2.3. BMC Isolation and Scaffold Seeding

The isolation of BMC was performed from syngeneic rats and followed the protocol of Assmus et al. [25] and as it was performed in their own prior studies [16,17,26]. Twenty donor animals were first given an overdose of pentobarbital (500 mg/kg intraperitoneally) and then killed by intracardiac blood sampling. This was followed by the removal of both the tibiae and femora. The bones were stored in a penicillin/streptomycin solution until further processing. For bone marrow harvesting, the condyles were removed using a side cutter. The bone marrow was then flushed into a sterile Petri dish using a disposable syringe (PBS (−/−) + 1% antibiotics). A homogeneous suspension was obtained by multiple pipetting. BMC were isolated by Ficoll density gradient centrifugation (1.077 g/cm^3^, Biochrom, Berlin, Germany). For this, BMC were diluted 1:3 with PBS (−/−), carefully layered onto the Ficoll, and centrifuged at 800× *g* and 20 °C for 30 min.

Subsequently, each BMC preparation was washed with 25 mL PBS at 350 g for 7 min. After determining the cell number, the colonization of the DBM scaffolds with the freshly isolated BMC was performed as described previously [19]. The sterile DBM scaffolds, pre-portioned by the manufacturer (German Institute for Cell and Tissue Replacement, Berlin, Germany), were sterilely transferred into one well (2.0 cm^2^) of each 24-well plate. This was followed by dropwise coating with 340 µL of BMC cell suspension (2.8 × 10^3^ cells/µL). After 5 min of incubation at 37 °C, the cell suspension that was not absorbed by the scaffold material was pipetted onto the material again. After further incubation (5 min, 37 °C), the cell loading procedure was complete. Scaffold cell colonization was verified by the visualization of 5(6)-carboxyfluorescein 3′,6′-diacetate N-succinimidyl ester (CFSE) pre-stained BMC seeded onto the scaffolds in a parallel setup using fluorescence microscopy.

The decellularized dermis (Epiflex, German Institute for Cell and Tissue Replacement, Berlin, Germany) was cut into 1.8 × 1.2 cm strips using a template before seeding. One strip per well was placed in a 6-well plate with the non-porous side up and overlaid with 1 mL PBS (+/+) and incubated at 20 °C for 10 min. BMCs were seeded to the membrane by applying a dynamic seeding procedure. In brief, BMC suspension (2 × 10^6^/mL) was dripped in three individual linearly arranged droplets (each 30 µL), covering the length of the membrane stripe, followed by five centrifugation steps at 300× *g* for each 1 min (Figure 1). Then, the membrane was inverted, and BMC suspension was dripped on the membrane, as described, followed by centrifugation for 1 min at 300× *g*. The BMC-loaded membrane was stored in PBS under sterile conditions at 37 °C until use.

### 2.4. Surgical Procedure

The surgery on the animals was performed analogously to the already established method in our laboratory [19]. The animals were anesthetized by the intraperitoneal injection of ketamine, Rompun, and midazolam. After checking the depth of anesthesia, the animals were weighed, and the right thigh of the animals was shaved and then disinfected. All further procedures now took place under sterile conditions. After the skin incision, the femur was bluntly dissected and visualized. The internal fixator, a 5-hole plate (Miniplate Lockingplate LCP Compact Hand 1.5 straight, DePuy- Synthes, Dubendorf, Switzerland), was placed in an adequate position on the femur and fixed with four 1.3 mm angular stable cortical screws (Compact Hand, DePuy-Synthes). The central hole was left free. A 5 mm bone defect was then sawed in the area of the unoccupied plate hole using a Gigli saw (RI-Systems, Landquart, Switzerland). Now the defect was wrapped with the 1.2 × 1.8 cm acellular dermis graft. Depending on the group, this was either dermis colonized with BMC or native dermis hydrated with PBS. The cavity created in the area of the defect was then filled with syngeneic cancellous bone, BMC-populated/untreated DBM granules, or BMC-populated/untreated DBM fibers, according to Table 1. The wound was closed in two layers with continuous subcutaneous stitches using a 4/0-monofilament nylon suture, and the animals were placed in their cages. The animals were examined daily for complications and behavioral changes. Analgesia (2.6 mg/kg Carprofen s.c. directly after surgery) was administered postoperatively via drinking water (Tramadol, 2.5 mg/100 mL) for 7 postoperative days. The animals were euthanized under inhalation anesthesia and intracardiac pentobarbital injection (500 mg/kg body weight). The femora were stored at −80 °C after the removal of the plate. Analyses were performed on the specimens in the following sequence: µCT analysis, biomechanical analysis, and (immuno)histological analysis. This allowed the sample material to be used optimally, and the number of animals needed was significantly reduced [27].

### 2.5. µCT (Microcomputertomographie)

Bone density and the callus volume of the harvested femora were measured using a high-resolution in vivo µCT Skyscan 1176 (Bruker AXS, Karlsruhe, Germany) (filter: Al 0.5 mm; rotation step: 0.5; rotation degree: 180; current: 500 µA; image average: 7; pixel: 18). The long axis of the femur was oriented orthogonally (perpendicular) to the X-ray beam path. The isotropic voxel size was 18 µm^3^. Two-dimensional CT images were scanned, reconstructed using a standard refolding procedure, and stored in 3D arrays. After µCT scanning, the femora were stored in 70% ethanol. To determine the bone mineral density in the defect area, a phantom with a known density was measured in parallel. For the calculation of the total volume and bone volume, a global threshold (60 to 240 gray levels) was defined in the respective defect volume. For this purpose, a virtual cylinder of 5 mm in length was placed centrally in the middle of the defect, and values for the total volume and bone volume were calculated using ImageJ (https://imagej.nih.gov/ij/notes.html, accessed on 24 October 2021). Due to the high radiological similarity between the syngeneic grafted bone and newly formed, already calcified bone tissue, and between DBM and not yet fully mineralized bone tissue, the newly formed bone tissue could not be reliably detected compared to previous studies [10]. Therefore, the total volume and the bone volume of the defect were determined. The bone mineral density (BMD) was then calculated from these data.

### 2.6. Mechanical Testing

The bones were stored in 70% alcohol until measurement. The bones previously examined in the µCt were also used to study the mechanical stability of the bone defect zone. The respective contralateral femora were used as reference bones. Stability testing was performed using a 3-point bending test and a material testing machine (Zwickline Z5.0, Zwick Roell, Ulm, Germany). The femora were always loaded in the same orientation with a wedge-shaped punch in the area of the defect zone. Force and bending were measured continuously. For evaluation, the Testexpert-II software (Zwick-Roell) was used to determine the relative ultimate load, as well as the relative bending stiffness of the operated bones compared to the healthy contralateral femur. The ultimate load was defined as a rapid force drop of at least 50%; the bending stiffness, in accordance with the literature, ‘was calculated from the slope of the linear elastic portion of the (load–deformation) curve’ [28]. Subsequently, the bones were again preserved in 70% alcohol.

### 2.7. Histology

The detection of BMC adherence to DBM was performed in a parallel approach using a pre-stained BMC followed by the qualitative detection of the cells by fluorescence microscopy. To confirm the adherence of BMC on DBM, prestaining with CFSE (5(6)-carboxyfluorescein 3′,6′-diacetate N-succinimidyl ester) and detection by fluorescence microscopy was performed as described previously [19].

The formation and the maturation of new bone tissue in the defect area were analyzed histomorphometrically on the basis of Movat pentachrome-stained, decalcified histological section preparations, as previously described [19]. The bone preparations were fixed in Zinc-Formal-Fixx (10%, Thermo Electron, Pittsburgh, PA, USA) for 20 h. Subsequently, the femora were decalcified over a period of 14 days in an aqueous solution containing 10 % EDTA (Sigma-Aldrich, Taufkirchen, Germany) and a 0.25 M Trizma base (Sigma-Aldrich, Taufkirchen, Germany). The pH of the solution was set to 7.4. After embedding the decalcified bones in paraffin, longitudinal sectional preparations (3 µm thickness) were made. Movat pentachrome staining was performed according to Garvey et al. [29] using a staining kit (Morphisto, Frankfurt, Germany) and following the manufacturer’s instructions. The percentage of bone tissue and cartilage tissue was calculated in relation to the defect area using ImageJ software (V 1.53 g, U. S. National Institutes of Health, Bethesda, MD, USA). The histological preparations were analyzed by light microscopy. High-resolution panoramic images of the entire defect area were obtained by automatically combining individual images using the BZ-9000 microscope (Keyence, Neu-Isenburg, Germany) and the BZII Analyzer Software (Keyence, Neu-Isenburg, Germany). The images were analyzed by two independent examiners who were blind to their group assignment. A bone healing score was used to assess the extent of bone defect healing and was performed according to Han et al. on Movat pentachrome-stained histological samples [30]. Improved bone healing was characterized by higher values.

### 2.8. SEM

The decellularized dermis was loaded with BMC following the dynamic seeding protocol described. This was followed by preparation for scanning electron microscopy analysis. For this purpose, the cell-loaded preparations were fixed in a 1% glutardialdehyde solution for 30 min and subsequently dehydrated in an ascending alcohol series (25%, 50%, 50%, 96%, 100%, 5 min each). Finally, the samples were incubated overnight with 1,1,1,3,3,3-Hexamethydilsilazane (Sigma-Aldrich, Steinheim, Germany). For scanning electron microscopy, the samples were vapor-deposited with gold 5 times for 1 min each (Agar Sputter Coater, Agar Scientific Ltd., UK) Sputter model type) and analyzed in a Hitachi scanning electron microscope (FE-SEM S4500, Hitachi, Dusseldorf, Germany) with a voltage of 5 kV.

### 2.9. Statistics

Group comparisons were performed using the nonparametric Kruskal–Wallis test with Bonferroni–Holm-corrected multiple Conover–Iman post hoc analysis. The results are presented either as boxplots or in the text as means and standard deviations. *p* values < 0.05 indicate a statistically significant difference. *p* values between 0.05 and 0.1 were considered a statistical trend.

## 3. Results

### 3.1. Animal Care and Complications

Of the 100 animals operated on, 12 animals could not be included in the evaluation. Six animals were excluded due to a peri-implant infection with pus formation, and three animals died directly after anesthesia, while three animals died during surgery, presumably due to femoral artery injury. The grafts were considered safe for use in previous experiments. There were no macroscopic side effects [19,27].

### 3.2. Cell Seeding and Seeding Efficacy

Cell colonization on DCD and DBM was examined by fluorescence and electron microscopy. Homogeneous distribution of cells on the membrane as well as on the DBM was observed (Figure 2a–c). Seeding efficiency was determined by the cells remaining in the well after seeding. After application, however, it became apparent that almost the entire cell suspension was absorbed by the dry, highly porous material (DCD and DBM). No relevant amount, in terms of the supernatant, could be isolated. Thus, the amount of liquid absorbed was more than 95%. A microscopic examination of the remaining liquid film showed only a few cells, so cell adherence can also be assumed to be more than 95%.

### 3.3. Bone Formation—Bone Score and Bone Mineral Density

The µCT examinations showed an increasing osseous bridging in all the extracted bones. The dermis wrapped around the defect gained a high mineral content, as described previously [24]. The dermis was clearly visible in all samples as a milky background in the defect gap and could not clearly be distinguished from the periosteal cover layer at the edges (Figure 3a–j). Almost complete consolidation could be seen in the DCD and the DCD + BMC group with SCB filling (Figure 3a,b). In the BMC-free approach, large, interconnecting compacta islands (Figure 3a, arrow) could be seen. These were not visible when BMC was added to the membrane (DCD + BMC) despite bony bridging Figure 3b). Filling with g-DBM also led to compact bridging with multiple compacta islands (Figure 3c). However, the compacta islands are small and only just merge. Interestingly, almost no compacta islands are seen in the DCD + BMC approach. Rather, bridging compact strands pass through the defect gap (Figure 3d). These strands are partially attached to the defect margins but also lie autonomously in the defect gap (Figure 3d, arrow). The addition of BMC to the g-DBM in DCD coating leads to significant bony bridging without visible compact islands and a defect gap that was hardly visible (Figure 3e). BMC in the coating continued to lead to bone formation in the sense of compacta strands and isolated islands. They were increasingly concentrated in the marginal zones. The defect gap was still clearly visible (Figure 3f). The use of fibrous DBM led to bone formation, which started leading from the marginal zones (Figure 3g–j) with only isolated small compacta islands. Regardless of BMC use in membrane or filling, the fracture gap was still visible (Figure 3g–j). Trabecular bone formation was seen primarily at the edges of the defect. Within the defect, compact bone was almost invariably seen at the time of examination (Figure 3a–j).

In a direct comparison of the individual fillings, both groups (DCD and DCD + BMC) showed a significantly lower bone mineral density (calculated from bone volume/total volume ratio (Table 2)) for the g-DBM groups compared to the filling with syngeneic cancellous bone (Figure 4a,b). In contrast, there was no significant difference for the f-DBM or f-DBM + BMC fillings (Figure 4a,b). The direct comparison of the DCD envelope with and without BMC showed no significant difference in bone mineral density (BMC) for all the fillings used (Figure 4c–g).

The histological sections reflect the impression of the µCT 3D reconstructions at the microscopic level. When the defect was filled with SCB, bony bridging was visible (Figure 5a,b). g-DBM-filled defects clearly show bone substance in the sense of multiple compacta islands in the defect gap (Figure 5c–f). When DCD was used, the compacta islands appeared confluent. In the DCD + BMC group, more isolated compacta islands were visible (Figure 5d,f). In all four approaches, the defect gap could still be delineated. The filling of the defect gap with f-DBM or f-DBM + BMC showed strand formations of the bone matrix (Figure 5g–j). Collagen strands could be seen between the bone substance. Interestingly, the subjective impression was reflected in the quantitative bone portion (Figure 5k,m). In the DCD group was no significant difference between the individual fillings in terms of bone and cartilage portion. For the DCD + BMC group, the distribution of bone tissue was partially inverted (Figure 5m). Here, SCB filling and f-DBM + BMC showed a significantly lower bone portion than the other groups. The cartilage portion showed no significant differences between the fillings (Figure 5n).

Overall, the bone healing score between the groups was at a similar level; therefore, existing differences could not be statistically validated. The highest score values were achieved in the defect filling of SCB without BMC with 22 points (Table 3).

### 3.4. Mechanical Resistance

The mechanical resistance of the extracted femora was determined in a 3-point bending test. In the DCD group, the filling with SCB showed a significantly higher breaking load than the filling with g-DBM + BMC, f-DBM, and f-DBM + BMC (Table 4; Figure 6a). Only the g-DBM filling did not differ significantly (a). Additionally, the bending stiffness was significantly higher for SCB than for g-DBM + BMC and f-DBM in the DCD group (Table 4; Figure 6b). For the filling with g-DBM and f-DBM + BMC, the difference was not significant. When the DCD was loaded with BMC, a change could be seen. Here, no significant differences were detected with respect to both the breaking load (Table 4; Figure 6c) and the bending stiffness (Table 4; Figure 6d). A direct comparison of the colonized and uncolonized DCD coating showed a significant difference with respect to the ultimate load only for the defect filling with f-DBM. Here, the DCD coating colonized with BMC exhibited a significantly increased ultimate load (Table 4; Figure 6).

## 4. Discussion

The masquelet technique is one of the “bandwagons” of critical-size bone defect treatment. A significant part of the research on this technique focuses on reducing the number of surgeries required [31]. Our group demonstrated that a decellularized dermis as a membrane substitute when using syngeneic cancellous bone as a bone void filler provided equivalent bone healing results [24]. This allowed the procedure to be reduced to a single surgery. The replacement of autogenous cancellous bone as a defect filler with artificial or allogeneic filling materials solves the major problem of harvest site morbidity. The results of a recent study showed equivalent bone healing when a demineralized bone matrix was used [19,27]. The aim of this work was to investigate the combination of DCD as a substitute for the induced membrane with DBM as the filler material—in both fiber and granular form. Syngeneic cancellous bone served as a control group for the defective filling. In addition, the influence of BMC colonization on both the membrane component and the filling was investigated.

The results indicate that DBM in its fibrous form may be superior to the granular form, especially when it is enriched with cells. There are essentially no significant differences in bone formation and stability compared to filling with syngeneic cancellous bone. The additional colonization of the dermis with BMCs does not appear to provide any additional benefit to bone healing in terms of BMD and bone maturation. However, ample evidence exists that BMCs also support bone healing. Hisatome et al. first described the effect of BMCs on bone healing in 2005. The group investigated whether BMCs improved neovascularization and bone healing in rabbit femoral defects [32]. Sun et al. also investigated the effect of BMCs on bone healing, focusing on vascularization [33]. In both studies, an improvement in vascularization and bone healing was observed. Moreover, using a rat model, Seebach et al. 2015 demonstrated that BMC-assisted bone healing provided comparable results to simultaneous treatment with an MSC/EPC mixture in terms of biomechanical stability, osseous remodeling, as well as new bone formation [16,17,34]. Compared to the cell-free control group (β-TCP only), there were consistently and significantly better results [17]. Nau et al., using the masquelet technique, demonstrated that defect filling with β-TCP in combination with either BMC or an EPC/MSC mixture supported the healing of bone defects at a similar extent to syngeneic bone [35]. In 2017, Janko et al. compared the seeding of human BMCs on three different scaffolds (DBM, bovine cancellous bone, β-TCP, and syngeneic cancellous bone as control) in the rat femoral defect model. Only low biomechanical stability was measured in combination with all three bone graft substitutes. However, the observation of bone maturation parameters, vascularization, callus formation as well as the osseous integration of the material showed that both cell-populated DBM and β-TCP were superior to bovine cancellous bone [6]. DBM was again superior to β-TCP and bovine cancellous bone in terms of the induction of numerous reparative genes in the defect zone. This suggested that the combination of DBM and BMCs was promising. The low biomechanical stability in Janko et al. compared with previous work was explained by the loss of the BMC effect due to the overnight storage of isolated cells for logistical reasons. Since, in the present study, the isolation of BMC was performed immediately before the operations were performed on the experimental day itself, this type of bias can be excluded here.

It was shown that the optimal concentration of BMCs was between the 2 × 106/mL and 1 × 10^7^/mL scaffold. Above the concentration of a 2 × 10^7^ BMC/mL scaffold, the positive effects that the BMCs had on new bone formation were regressive [33]. Since over 95% were absorbed, the cell concentration of the 2.8 × 10^6^ BMC/bone defect that we used in the present study was, therefore, within the range determined, although this did not take into account the surroundings of the bone defect with a membrane. On the one hand, this represented a mechanical barrier and, on the other hand, was partly and additionally occupied by BMCs. To what extent the DCD as a barrier increased the concentration of BMCs in the bone defect area beyond the optimal level—where the emigration of cells into neighboring tissues was prevented—has not been clarified. Therefore, the question is why increased BMC cell number concentration leads to a worsening of bone healing. A dose–response study by Janko et al. could not clarify this [33]. The hypothesis investigated there, that the foreign body reaction would be increased by the CD68-positive macrophages increasing proportionally to the concentration of BMCs, could not be substantiated. The concentration of CD68 positive macrophages was not dependent on the concentration of BMCs in the defect area.

Monocytes and macrophages are among the first cells recruited to injured tissue. In addition to their role in phagocytosis, antigen uptake and processing, and immune regulation, they perform multiple roles in initiating and regulating initial regenerative responses [36,37]. Monocytes in BMC are predominantly the pro-inflammatory subtype CD16 [26]. BMC therapy thus provides increased pro-inflammatory monocytes, although it is possible that the cells, similar to the migrated monocytes, undergo a switch to an anti-inflammatory subtype. Earlier studies suggest that the indirect osteoinductive effects of therapeutic cells, rather than their directly integrating into healing bone, are responsible for the beneficial effects of bone regeneration [38,39]. Actual unpublished data from our research group suggests a high concentration of cytotoxic CD8+ T cells in BMC. When CD8+ T-cells accumulate, the primarily beneficial inflammation necessary for bone healing could be elevated to a deleterious level. Schlundt et al. demonstrated that by improving the ratio of proinflammatory CD8+ T cells to regulatory CD4+ T-cells (in terms of decreasing CD8+ cells), bone healing improved [39]. Potentially, therefore, an increased proportion of CD8+ T-cells could have led to hyperinflammation and, thus, the reversal of the beneficial effect of BMC on bone healing.

The inflammatory phase of bone healing is characterized by the development of fracture hematoma. The hematoma contains granulocytes, lymphocytes, and monocytes, as well as a variety of cytokines and growth factors that are released by the migrated immune cells and platelets, such as TNF-α, IL-1β, IL-6, IL-8, IL-10, IL-12, BMP-2, -4, PDGF, TGF, and VEGF [40,41]. These factors contribute to the continuous recruitment of immune cells in the sense of a positive feedback loop. In some circumstances, the overregulation of the feedback loop may occur due to an excessive concentration of BMC. This could lead to a prolongation of the inflammatory phase, resulting in an increased formation of connective tissue structures in the defect and ultimately leading to restricted bony development up to the formation of pseudarthrosis. At the time of killing the test animals, the inflammation largely subsided [42]. Investigations at a leading time of bone healing could provide further insights [43]. The hypothesis was that the application method could lead to a limited formation of connective tissue structures in the defect and, ultimately, to the formation of pseudarthrosis.

The hypothesis that the shape of DBM could influence bone healing capacity is not new. However, in vivo studies investigating the effect of different modes of application on bone healing are rare. In addition to the prior work where we compared fiber DBM and granular DBM with the gold standard in the rat femoral defect model, Schouten et al. hypothesized in 2005 that crushed DBM had greater osteoinductive properties than non-crushed DBM via the increased combined surface area [27].

Although the DBM mineral amount during manufaction is reduced to 1–6 %, the remaining tissue shows microscopically a trabecular structure consisting of collagen (mainly type 1), non-collagenous proteins and growth factors [44]. These proteins are thought to serve as biological drivers for osteoconductive, and osteoinductive properties [7,44,45,46]. For example, Wildemann et al. identified BMP-2, TGF-β1, IGF-1, and VEGF as central regulators of bone metabolism and, thus, of bone healing and remodeling [7]. These mediators were incorporated into the bone by osteoblasts during initial bone formation. During remodeling, they were exposed to osteoclast activity which unfolded their effect. They also have a direct influence on the basic multicellular units (BMUs) [47]. This consists of osteoclasts resorbing bone, osteoblasts replacing bone, osteocytes within the bone matrix, bone lining cells covering the bone surface, and capillary blood supply. For example, BMP-2 has an osteoinductive effect on osteoblasts. This increase in activity was stimulated by TGF-β1 [48]. Collagen-1 has osteoconductive effects on osteoclasts and ensures their immigration. The proteins in the DBM are presumably more readily available or already exposed. Osteoinductive properties are attributed to growth factors, especially BMP-2, with decreasing calcium levels leading to a decomplexation of growth factors and proteins [45,49,50]. A stimulatory effect on the BMU can be assumed. It is not clear whether this results in osteoclast or osteoblast stimulation. The influence on the character of bone healing—enchondral or intramembranous bone healing—has also not yet been clarified. Since both processes occur in parallel in normal bone healing, both processes are to be expected under the influence of DBM [51]. It would be interesting if this led to a different balance of those processes. Furthermore, interconnected pores and the microstructured surface offer good conditions for cell adhesion, cell ingrowth, vascularization, and connective tissue proliferation (osteoconductivity) [52].

However, this hypothesis could not be confirmed in the animal experiment conducted for this purpose [53]. Sanaei et al. found the superiority of tubular DBM over DBM chips on the bone defects of pigeons—however, they used avian DBM prepared by the research group [54]. The results of a recent study by the research group of Verboket et al., which compared syngeneic bone with fiber and granular DBM in a rat femoral defect model, demonstrated the equivalence of DBM fiber and syngeneic bone in terms of biomechanical stability, while DBM granules performed significantly worse by comparison [27]. In addition to the potentially improved release of osteoinductive proteins from the fibrous material, an observation from bone substitute research may also provide an explanation. It was shown there that osteoconduction could be significantly improved with structures of 10–100 µm. As a result, cells can migrate and adhere more strongly [55,56]. The additional colonization of the dermis with BMCs does not seem to provide any additional benefit for bone healing in terms of BMD and bone maturation. This supports the thesis for membrane function. Injected bone material is limited by the membrane and remains in the defect area; furthermore, its resorption is prevented and the ingrowth of connective tissue is inhibited, which in the further course could promote the development of pseudarthrosis. This assumption is supported by the results of Tarchala et al. They were able to show that the use of a non-biological polytetrafluoroethylene membrane led to comparable new bone formation as with the induced membrane, according to masquelet [20,57]. As already mentioned, our study showed equivalent healing results when replacing the induced membrane with DCD [24]. Whether an improvement in the results could be achieved by the cell colonization of the membranes has not yet been attempted. The fact that the data collected here, with one exception, show no significant differences in c (BMD, Figure 4) between the native DCD coverage and cell-populated DCD coverage groups is further evidence of the central barrier function. Only the breaking load was significantly increased when filled with f-DBM + BMC toward native f-DBM. The “additional” cells on the membrane did not cause any significant improvement. It did not matter that a fraction of the cells was applied to the inner side of the membrane. Bone growth lead from the fracture ends (Figure 5) into the defect zone. Basically, this shows that the colonization of the membrane has no leading influence on the actual bone healing of the defect but works as a barrier that keeps the infilled bone graft material in place. Furthermore, by limiting the range of motion of the filler material, a stable defective envelope could help to protect any formed bony connections. Decades of experience in oral and maxillofacial surgery further support the hypothesis that the ingrowth of cells from the surrounding soft tissue mantle leads to the formation of pseudarthrosis and can be prevented by the implantation of a membrane covering the defect [58].

## 5. Conclusions

This study also showed the advantage of fibrous DBM with equivalent healing results compared to syngeneic cancellous bone. The addition of cells, on the other hand, did not provide any significant advantage in context with the here used bone substitutes and the used membrane. Future investigations should focus on the optimization of the filling material.

## Figures and Tables

**Figure 1 cells-12-01289-f001:**
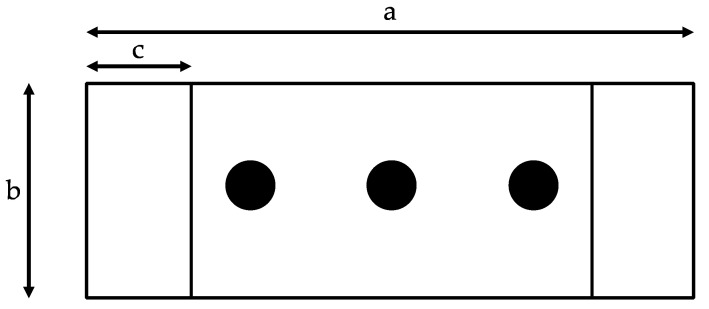
Schematic drawing of an acellular dermis piece. a: 1.8 cm; b: 1.2 cm; c: 0.2 cm. The dots denote the site where cell suspension was dropped. The separated marginal areas are suture areas.

**Figure 2 cells-12-01289-f002:**
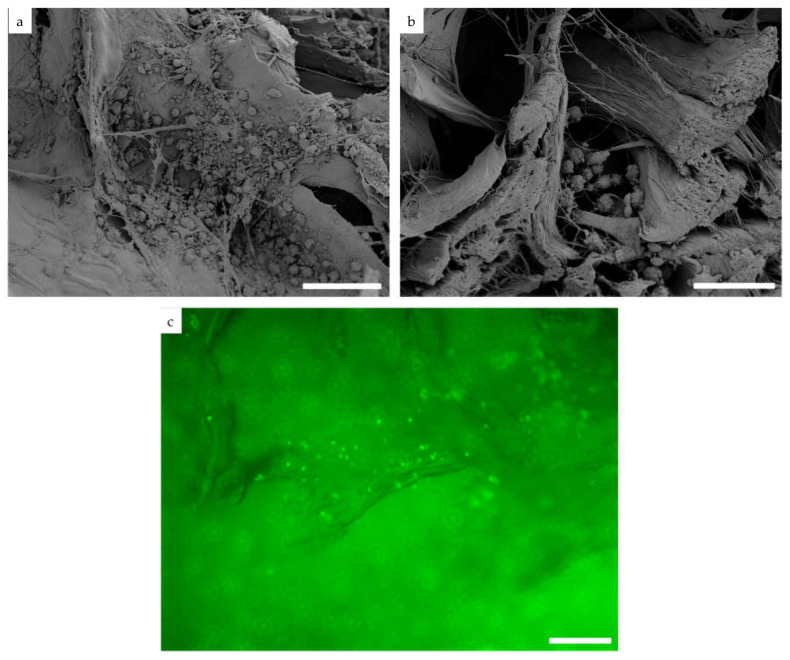
Detection of seeded BMC on the surface (**a**) or inside the DCD (**b**) using a scanning electron microscope. Detection of CFSE pre-stained BMC (green dots) (**c**) after seeding on f-DBM using fluorescence microscopy. Bar represents 20 µm.

**Figure 3 cells-12-01289-f003:**
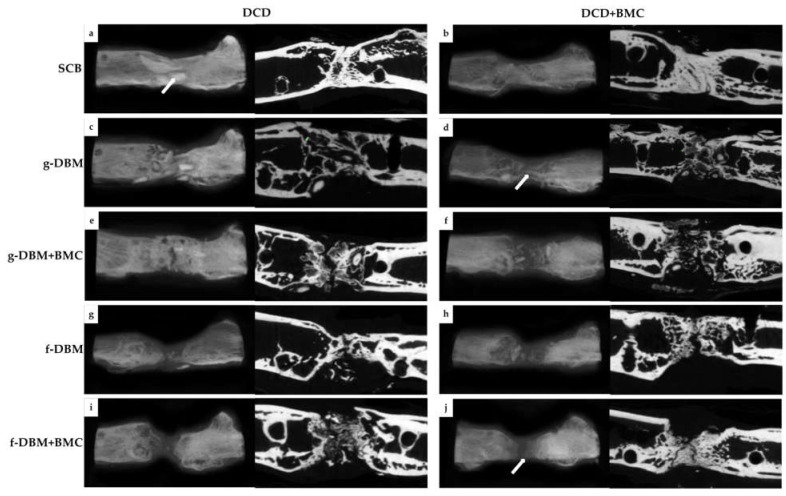
Three-dimensional reconstructions and sectional images of the µCT scans. There was an increasing consolidation of all defects after 8 weeks. Almost complete consolidation with compacta islands ((**a**), arrow) was evident in the SCB group (**a**,**b**). The fracture gap was completely bridged. The g-DBM group (**c**–**f**) also showed solid bridging of the defect gap when using the cell-free membrane (DCD), with only a barely demarcated fracture gap. The use of BMC-populated membrane led to a lower degree of bridging (**d**,**f**). Here, mainly compacta strands were impressed ((**d**), arrow). Complete bridging had not yet occurred. In the f-DBM groups (**g**–**j**), much less progress was seen. The fracture gap was still clearly delineated. Here, the ingrowth of marginal bone was leading ((**j**), arrow). Trabecular bone formation was seen primarily at the edges of the defect. Within the defect, compact bone was almost invariably seen at the time of examination.

**Figure 4 cells-12-01289-f004:**
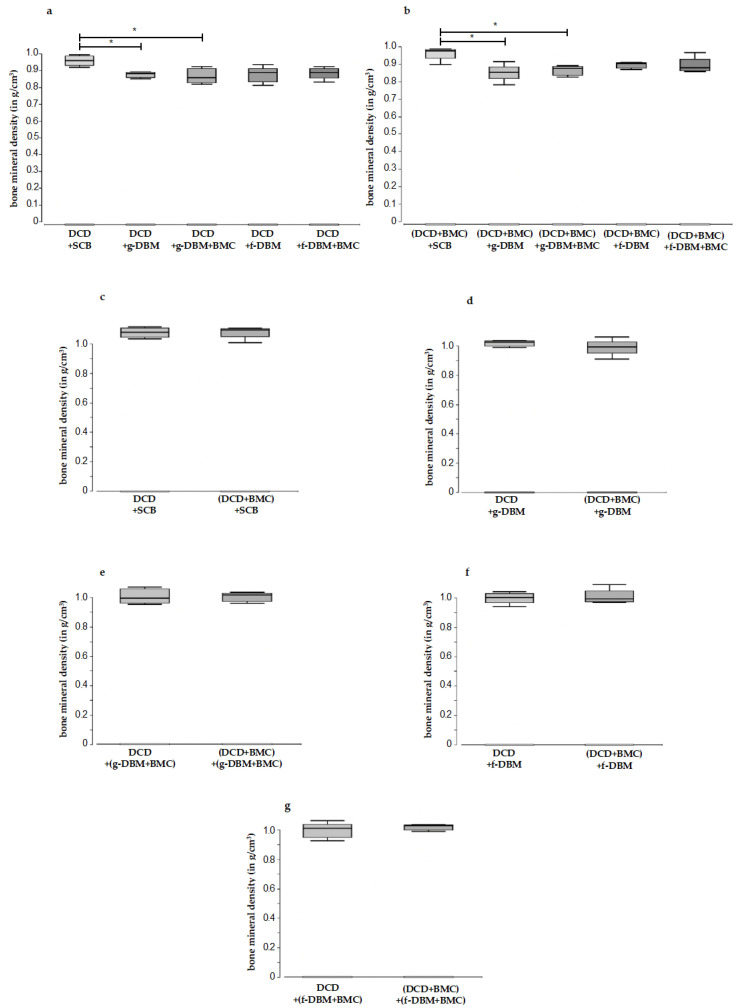
Comparison of bone mineral density (g/cm^3^) in the defect region for different defect fillings (**a**,**b**) and different DCD coatings (with and without BMC) (**c**–**g**). In both groups (DCD without and with BMC), significantly greater bone mineral density could be detected for syngen bone filling towards g-DBM filling. There was no significant difference in bone mineral density in comparison to filling with f-DBM (without and with BMC) towards syngen bone filling. (DCD: decellularized membrane; BMC: Bone mononuclear cells; f-DBM: fibrous demineralized bone matrix; g-DBM: cancellous demineralized bone matrix). No significant difference could be detected between BMC-colonized and uncolonized DCD with different fillings (DCD: decellularized membrane; BMC: Bone mononuclear cells; f-DBM: fibrous demineralized bone matrix; g-DBM: cancellous demineralized bone matrix). “*” marks statistical significance (*p* < 0.05).

**Figure 5 cells-12-01289-f005:**
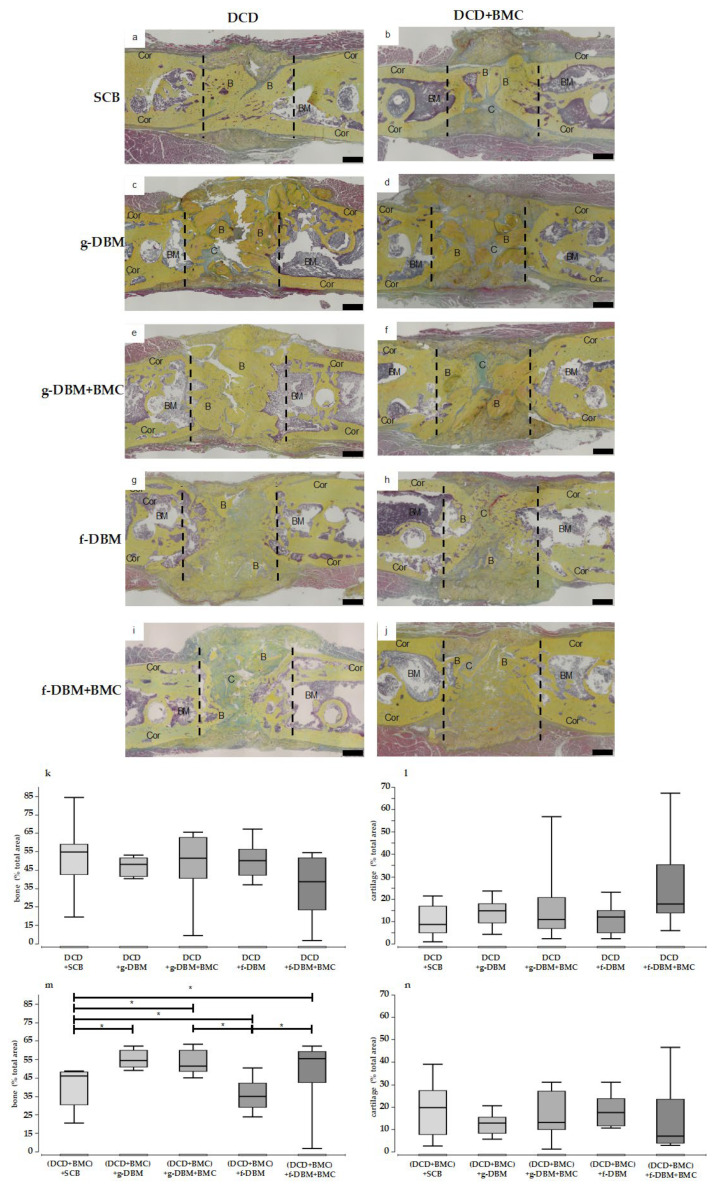
Movat-pentachrome staining of the defect region. DCD + SCB (**a**), DCD + g-DBM (**c**), DCD + (g-DBM + BMC) (**e**), DCD + f-DBM (**g**), DCD + (f-DBM + BMC) (**i**) and DCD with BMC addition: (DCD + BMC) + SCB (**b**), (DCD + BMC) + g-DBM (**d**), (DCD + BMC) + (g-DBM + BMC) (**f**), (DCD + BMC) + f-DBM (**h**), (DCD + BMC) + (f-DBM + BMC) (**j**). Bone tissue appears yellow, and cartilage greenish. Histology: Size Bar represents 1 mm, dotted line defect zone, BM = bone marrow, Cor = corticalis, C = cartilage, B = Bone. Corresponding relative quantification in percentage of bone portion (**k**,**m**) and percentage of cartilage portion (**l**,**n**). There was no significant difference in bone or cartilage portion for the pure DCD group. In the DCD + BMC group (**m**), bone portion was significantly decreased with SCB and f-DBM. Cartilage portion did not show any significant differences in this group (**n**). “*” marks statistical significance (*p* < 0.05).

**Figure 6 cells-12-01289-f006:**
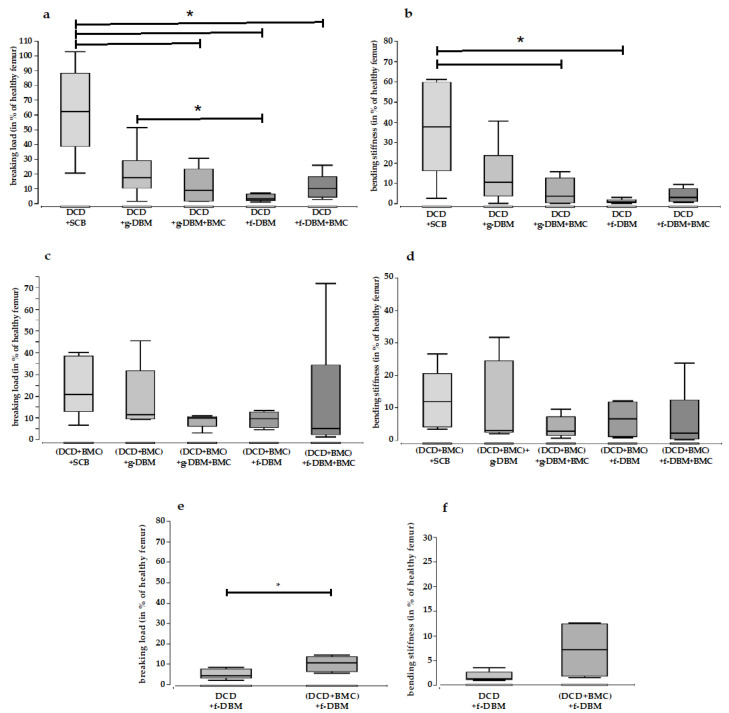
Breaking load and bending stiffness in % of healthy femur. In the decellularized dermis group (DCD) without BMC (**a**), a significantly higher breaking load was observed when the defect zone was filled with syngeneic cancellous bone (SCB) compared to all other fillings. An exception was the g-DBM filling with no significant difference. The investigation of the bending stiffness for the DCM group without cells (**b**) with SCB filling showed significantly higher values compared to g-DBM + BMC and f-DBM. In the DCD group with BMC (**c**,**d**), no significant difference could be seen between the fills for ultimate load and bending stiffness. A direct comparison of colonized and uncolonized DCD coating showed a significant difference with respect to the breaking load only for the defect filling with f-DBM (**e**). Here, the DCD coating colonized with BMC exhibited a significantly increased ultimate load. This difference was not seen in the bending stiffness (**f**). “*” marks statistical significance (*p* < 0.05).

**Table 1 cells-12-01289-t001:** Groups of different material and cell combinations and number of animals assigned to each group.

Material	Decellularized Dermis (DCD)	Decellularized Dermis + Bone Mononuclear Cells (DCD + BMC)
Syngenic cancellous Bone (SCB)	*n* = 10 (group 1)	*n* = 10 (group 2)
DBM granules (GDBM)	*n* = 10 (group 3)	*n* = 10 (group 4)
DBM granules + BMC (GDBM +BMC)	*n* = 10 (group 5)	*n* = 10 (group 6)
fibrous demineralized bone matrix (f-DBM)	*n* = 10 (group 7)	*n* = 10 (group 8)
fibrous demineralized bonematrix + BMC (f-DBM + BMC)	*n* = 10 (group 9)	*n* = 10 (group 10)
sum	50	50

**Table 2 cells-12-01289-t002:** Bone volume/total volume in the experimental groups. Mean values and standard deviation in % is shown. No statistically significant differences were found.

Material	DCD	DCD + BMC
SCB	0.74 (±4.2%)	0.76 (±4.8%)
g-DBM	0.64 (±4.1%))	0.58 (±3.4%))
g-DBM + BMC	0.70 (±4.8%)	0.67 (±5.1%)
f-DBM	0.67 (±4.1%)	0.59 (±5.6%)
f-DBM + BMC	0.58 (±3.4%)	0.51 (±5.8%)

**Table 3 cells-12-01289-t003:** Bone healing scores. The bone healing score was determined based on Movat–Chrome staining according to Han et al. [30].

Material	DCD	DCD + BMC
Syngeneic cancellous Bone (SCB)	22	21
DBM granules (GDBM)	19	18
DBM granules + BMC (GDBM +BMC)	18	18
fibrous demineralized bone matrix (f-DBM)	21	20

**Table 4 cells-12-01289-t004:** Median breaking load and bending stiffness in % of healthy femur.

	Breaking Load % (Median)	Bending Stiffness % (Median)
DCD + SCB	62.1	27.8
DCD + g-DBM	17.4	10.5
DCD + g-DBM	8.5	3.4
DCD + (g-DBM + BMC)	3.1	0.6
DCD + (f-DBM + BMC)	10.0	2.7
(DCD + BMC) + SCB	20.8	11.8
(DCD + BMC) + g-DBM	11.2	3.0
(DCD + BMC) + (g-DBM + BMC)	9.8	2.7
(DCD + BMC) + f-DBM	9.6	6.5
(DCD + BMC) + (f-DBM + BMC)	5.1	2.1

## Data Availability

The data presented in this study are available on request from the corresponding author.

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
