# Peer review of "One Stage Masquelets Technique: Evaluation of Different Forms of Membrane Filling with and without Bone Marrow Mononuclear Cells (BMC) in Large Femoral Bone Defects in Rats"

_cells, 2023, doi:10.3390/cells12091289_

Round 1

Reviewer 1 Report

In this work, the authors developed one stage masquelets technique to clarify whether the SCB can be replaced by granular (g-DBM) or fibrous (f-DBM) demineralized bone matrix and whether colonization of the DCD and/or the DBM defect filling with bone marrow mononuclear cells (BMC) leads to improved bone healing. A critical femoral bone defect model in SD rats were used and subjected to histological, radiological, and biomechanical analysis. Despite the some effort that has been done by the authors, a number of questions are arising in course of manuscript reading. Please address following questions, to make the results and claims presented in the study more valid and conclusive.

1. Figure 3 presented the data of mechanical testing, but not found the results of movat-pentachrome staining of the defect region.

2. Some results should be presented in scatter plots, not floating bars.

3. The standard deviation of part of results, such as Figure 5c, was too high, which reduced credibility of the data.

4. Quantitative analysis was performed on tissue sections in this manuscript, but the original section images were not included. Please add the original section images.

5. How about the basic cytological experiments of BMC, such as CCK-8 and AM/PI?

6. Micro-CT was utilized to analyze the extent of bone growth, but parameters such as BV/TV and other analyses were not mentioned. Please provide related data.

7. How to calculate cell density on the DCD membrane after inoculation?

8. The role of inflammatory factors in bone healing was discussed in the discussion section, but the relevant results was not found in the manuscript. Please add it.

Author Response

See the file attached. 

Reviewer 2 Report

The authors described a new DCD-based approach for the treatment of femoral bone defects. Some points need to be clarified in the review.

In the "Materials and Methods" section, the authors described histology, but no results were shown in the "Results and Discussion" section. 

The manuscript mentions the use of micro-CT and mechanical testing, but there are no histologic or immunologic studies to evaluate tissue response to treatment. This is especially important when considering the use of a decellularized dermis membrane, as it could potentially elicit an immune response.

The authors also discussed immunologic/inflammatory responses with citations. It would be important to include their own data to demonstrate the compatibility of the DCD materials tested. 

The manuscript claims that surgery can be saved by using the decellularized dermis membrane, but it is unclear how this conclusion was demonstrated. The study didn't include a comparison group to evaluate whether the membrane actually improves outcomes compared to the classic two-stage Masquelet technique alone.

Author Response

See the file attached. 

Reviewer 3 Report

The paper entitled One stage Masquelets technique: Evaluation of Different Forms of Membrane Filling with and without Bone Marrow Mononuclear Cells (BMC) in Large Femoral Bone Defects in Rats is potentially interesting. The authors showed that showed emerging bony bridging of the defect zone by using granular (g-DBM) or fibrous (f-DBM) demineralized bone matrix in a large femoral bone defect rat model, while the addition of BMCs showed no further improvement in bone healing.  Stability and bone formation were comparable to the control group. In conclusion, demineralized bone matrix might serve as an agent for bone defect repair. The study is well presented.

There are some possible issues

Figure 2. 3D reconstructions of the μCT scans. Can you provide uCT parameters such as bone volume versus tissue volume with quantitative analyses?

Figure 2. 3D reconstructions of the μCT scans.  Can you provide histology to see if major cells such as osteoclasts, osteoblasts, and endothelial cells are affected? 

Previous studies (for example PMID: 22465238) have shown that bone marrow and bone microenvironment contain many cell types such as macrophages, osteoclasts, osteoblasts, bone lining cells, osteomacs, and vascular endothelial cells, etc. It would be informative to discuss how demineralized bone matrix might regulate cell types in the bone local environment by referring to this report, which could facilitate bone defect healing. 

Do you have any in vitro data on the effects of demineralized bone matrix on bone cells, such as osteoclasts, and osteoblasts? Or osteoinductive and osteoconduction effects?

In this paper, the authors mentioned possible osteoinductive and osteoconduction effects of the demineralized bone matrix with bone healing. The process of bone healing is regulated by intramembranous and endochondral ossification as previously reported (for example, PMID: 33385019). It would be informative to discuss the cellular mechanism of demineralized bone matrix-mediated bone healing processes from this report.

There are some possible typos

counterstaining with hematoxilin?? counterstaining with hematoxylin?

Syngenic cancellous Bone? Syngeneic cancellous Bone?

Author Response

See the file attached. 

Reviewer 4 Report

Review note

This paper evaluate the restoring potential of different forms of membrane filling for large femoral bone defects in rats. I appreciate the authors’ preclinical findings, which will help further clinical practice. However, to grow into a publication, I think there are some issues the authors need to address.

1.     The evaluation of BMD was based on uCT scans, thus, it would be more logical to put them together (for example, uCT images shown as in figure 1 and the BMD shown in figure 2).

2.     Since you have uCT scanned the samples, it would be easier for the readers to see the differences through 3-D visualization (trabecular and cortical bones).

3.     Similarly, it would be better to show the visualization of the Movat-pentachrome staining of the defect region besides quatifications.

4.     It is important to exhibit the SD or SEM through appropriate statistical analysis in he tables. The median or everage score does not completely reflects the discrepancy between groups.

5.     Minor concern: The abbreviation “DBM” is not introduced in the abstract.

In summary, I feel the study is interesting and clearcut. However, improvement is needed. I hope the author(s) could find some of the above discussions helpful for improving the paper. 

Author Response

See file attached.

Round 2

Reviewer 1 Report

The author has made targeted revisions to the previous round of review comments, and this version is currently suitable for publication.

It is recommended to make appropriate grammar modifications for formal publication.

Reviewer 2 Report

questions from round one are addressed.

Reviewer 3 Report

the authors have addressed questions and the paper has been improved.